# TARGETED MODEL INVERSION: DISTILLING STYLE ENCODED IN PREDICTIONS

## ABSTRACT

Previous model inversion (MI) research has demonstrated the feasibility of reconstructing images representative of specific classes, inadvertently revealing additional feature information. However, there are still two remaining challenges for practical black-box MI: (1) reconstructing a high-quality input image tailored to the observed prediction vector, and (2) minimizing the number of queries made to the target model. We introduce a practical black-box MI attack called **T**argeted **M**odel **I**nversion (TMI). Our approach involves altering the mapping network in StyleGAN, so that it can take an observed prediction vector and transform it into a StyleGAN latent representation, which serves as the initial data point for subsequent MI steps. Later, TMI leverages a surrogate model that is also derived from StyleGAN to guide instance-specific MI by optimizing the latent representation. These mapping and surrogate networks work together to conduct high-fidelity MI while significantly decreasing the number of necessary queries. Our experiments demonstrate that TMI outperforms state-of-the-art MI methods, demonstrating a new upper bound on the susceptibility to black-box MI attacks.

## 1 INTRODUCTION

MI refers to an adversarial attack that reconstructs training data or class-representative instances based on the output from a target machine learning (ML) model. Assuming an adversary who is able to eavesdrop or obtain an output prediction from a target model, successful MI attacks involve reconstructing an input image corresponding to that specific output or generating a representative image belonging to the same class that the prediction indicates. Consequently, these reconstructed images expose privacy-sensitive features that the model owners or its users did not anticipate revealing through the output predictions. Prior studies have vastly investigated the feasibility and efficacy of MI against deep neural networks (DNNs) (Fredrikson et al., 2015; He et al., 2019). Recently, Yang et al. (2019) proposed a training-based attack that utilizes a DNN generator, enabling it to reconstruct an image based on a given prediction vector. Subsequent works focused on improving the fidelity of reconstructed images by adopting generative adversarial networks (GANs) (Zhang et al., 2020b; Chen et al., 2021; Kahla et al., 2022; Yuan et al., 2023; Han et al., 2023) or StyleGANs (Wang et al., 2021; An et al., 2022; Struppek et al., 2022).

We posit that there still remains large room for improvement in conducting practical black-box MI attacks. Specifically, we propose two key challenges to overcome: (1) minimizing the number of necessary queries to a target model and (2) enabling instance-specific reconstruction. Numerous studies have assumed a strong white-box adversary who is able to access target model parameters, thereby leveraging gradients in performing MI (Fredrikson et al., 2015; Zhang et al., 2020b; Wang et al., 2021; An et al., 2022; Struppek et al., 2022). Moreover, existing black-box MI attacks (Yang et al., 2019; Kahla et al., 2022; Han et al., 2023) require an excessive number of queries for a target model, rendering them impractical. For example, An et al. (2022) required 160k queries to reconstruct a single image. Furthermore, previous researchers have focused on reconstructing class-representative images rather than the original input images specific to the corresponding prediction vector. Class-representative images often omit intra-class differences within their class, which undermines the chances of reconstructing privacy-sensitive features. For instance, when a target task for MI is gender classification, class-representative images display a generic female face, not a specific woman involved in training (Melis et al., 2019). The difficulty is even exacerbated when a target task involves

large variances in each class. For example. we observed that, with the NIH Chest X-ray dataset (Wang et al., 2017), previous methods are unable to reconstruct task-agnostic features such as gender or age.

To tackle the aforementioned challenges, we introduce Targeted Model Inversion (TMI), a novel MI framework that performs instance-specific reconstruction while leveraging only a restricted set of black-box queries to a target model. TMI consists of two steps: preparation and inversion. In the preparation step, TMI employs a StyleGAN (Karras et al., 2020) network trained on a dataset in which the underlying distribution is similar, yet different from the training set of the target model. Then, its mapping and discriminator networks are modified to project the prediction vector to the StyleGAN latent space and to act as a surrogate model, respectively. These networks are trained using StyleGAN-generated images and their corresponding predictions from the target model, eliminating the need for an additional dataset. In the inversion step, TMI locates the initial StyleGAN latent corresponding to a target prediction vector observed from the target model by using the new mapping network. It then optimizes this latent to generate an image that prompts the surrogate model to emit a prediction vector similar to the target prediction vector.

The core idea of TMI is to construct a new mapping network that approximates a style latent corresponding to the target prediction vector, which is then further optimized through signals from the surrogate model. Both distilled from a benign StyleGAN network, these two modified components help significantly reduce the number of required queries for successful MI while conducting instance-specific inversion for a given prediction vector.

We evaluate TMI by comparing it with other state-of-the-art white-box and black-box MI attacks. We demonstrate the superiority of TMI even with a much smaller query budget; TMI attains 0.2067 in terms of class-wise coverage, significantly surpassing the performance of existing state-of-the-art MI attacks. These experiment results demonstrate that the TMI attack enables practical black-box MI with high fidelity without requiring white-box access to a target model.

## 2    MODEL INVERSION ATTACK

An MI attack refers to an adversarial attempt to reconstruct an input image $x \in \mathcal{X}$ based on the target output prediction $\hat{y}_t \in \mathcal{Y}$ obtained from a target classifier $f : \mathcal{X} \mapsto \mathcal{Y}$. The reconstructed image $x'$ may inadvertently leak privacy-sensitive features that were never expected by the model owner or its users. Formally, the adversary's objective is to compute an inversion image $x'$ satisfying the following equation:

$$x' = \arg\min_{x \in \mathcal{X}} \mathcal{L}_{pred}(f(x), \hat{y}_t) \tag{1}$$

with a loss function $\mathcal{L}_{pred}$ (e.g., cross-entropy loss or $\ell_2$ loss) that quantifies the dissimilarity between the observed prediction vector $\hat{y}_t$ and the target model output $f(x)$.

To overcome the challenge of reconstructing high-fidelity images in an input space (i.e., $\mathbb{R}^{3 \times 224^2}$) based on a prediction vector in a limited model output space (i.e., $\mathbb{R}^{\mathcal{K}}$, where $\mathcal{K}$ refers to the number of classes in $f$), previous researches have explored different attack methods. Early MI studies focused on reconstructing low-resolution grayscale facial images or simple datasets like MNIST. For instance, Fredrikson et al. (2015) applied an analytic method of finding $x'$ in Equation 1. Later, Yang et al. (2019) proposed using a dedicated neural network consisting of multiple transposed convolutional layers that directly map the observed prediction vectors onto the input space, expanding the attack vector of MI to relatively complicated neural networks. However, their attack focused on reconstructing low-resolution gray-scale input images.

To further improve MI, subsequent studies have proposed leveraging the GAN generators (Zhang et al., 2020b; Chen et al., 2021; Wang et al., 2021; An et al., 2022; Struppek et al., 2022; Kahla et al., 2022; Yuan et al., 2023; Han et al., 2023). The generator $g : \mathcal{Z} \mapsto \mathcal{X}$ operates as an image prior, generating input images in $\mathcal{X}$ from Gaussian latent vectors in $\mathcal{Z}$. Instead of directly optimizing in the input image space $\mathcal{X}$ as in Equation 1, GAN-based approaches perform optimization within a more constrained space $\mathcal{Z}$. Recent MI researchers have adopted StyleGANs (Karras et al., 2019; 2020) to attain higher-fidelity reconstruction (An et al., 2022; Struppek et al., 2022); they perform optimization in a newly introduced intermediate latent space $\mathcal{W}$.

Note that the optimization process in MI typically requires computing gradients using the target model, thereby assuming the presence of a white-box adversary who is able to access the target

model parameters. Follow-up studies have proposed attack methods to simulate the optimization using only black-box queries. These attack techniques include genetic algorithms (An et al., 2022), decision boundary estimation (Kahla et al., 2022), and reinforcement learning (Han et al., 2023) as proxies for the optimization process. Although achieving state-of-the-art performance compared to traditional black-box approaches, we argue that all the existing methods still fail to address two following challenges: practicality and instance-specific inversion.

**Practicality.** Existing black-box MI methods still demand a prohibitively large number of queries to the target model. This poses practical challenges, particularly when considering the limitations imposed by Machine Learning as a Service (MLaaS) providers. These providers often enforce rate limits on API calls, restricting the number of queries (e.g., Clarifai - 5000/day, DatumBox - 1000/day). The requirement for a high volume of queries not only tampers with the practicality of MI but also raises the risk of attack detection because an abnormal number of queries can potentially be flagged.

**Instance-specific inversion.** Test-time MI attacks can be categorized into two groups (Yang et al., 2019): instance-specific MI and class-representative MI. The former refers to a scenario in which the attacker infers a victim's input instance for an observed prediction output. On the other hand, class-representative MI focuses on reconstructing generic images for a single output class in a target model. Whereas a larger volume of previous research focused on conducting class-generic MI that reveals class-bound features (Fredrikson et al., 2015; Yang et al., 2019; Zhang et al., 2020b; Chen et al., 2021; Wang et al., 2021; An et al., 2022; Struppek et al., 2022; Kahla et al., 2022; Yuan et al., 2023; Han et al., 2023), instance-specific MI has been largely understudied. Due to the difficulty of instance-specific MI that requires the reconstruction of subtle and instance-specific image features, it was deemed possible under specific conditions, such as the collaborative inference setting, where intermediate representations and gradients are accessible to the adversary (Melis et al., 2019; He et al., 2019; 2020). For instance, class-generic MI reveals only class-bound features, including race, gender, and age, in facial recognition tasks. On the other hand, the instance-specific MI seek reconstruction of additional instance-specific features, such as accessories, facial expressions, or posture, as well as the class-bound features.

## 3 THREAT MODEL

We assume a target classifier $f$, providing black-box access where the adversary (Eve) is able to query an input image $x$ to obtain $\hat{y}$, where $\hat{y}$ is the corresponding output prediction in the form of a confidence vector. The designed goal of TMI is to reconstruct the specific input $x$ that produced the target prediction $\hat{y}_t$. In general, one does not anticipate an output prediction to convey subtle details of the corresponding input, so they are regarded less confidential compared to the input data itself. This trend is evident in regulations like HIPAA, where the guidelines for storage and transmission of medical images are more strict than the rules regarding diagnostic predictions (Moore & Frye, 2019). Furthermore, in the field of confidential computing that protects the privacy of user input to a cloud-provided ML service, the prediction outputs are excluded from encryption, allowing direct access from cloud providers with malicious intents (Gu et al., 2018; Narra et al., 2019). Accordingly, $\hat{y}_t$ is often leaked, eavesdropped, forged, or carelessly exposed to cloud service providers or man-in-the-middle adversaries in real-world scenarios. These exemplary scenarios also include users posting their prediction results on social media (Yang et al., 2019), split inference settings where the inference result is sent to different parties (He et al., 2019), or medical professionals sharing diagnosis predictions for educational or consultative purposes.

Eve leverages an auxiliary dataset $\mathcal{D}_{aux}$ of which the underlying distribution is similar to those of the original dataset $\mathcal{D}$ upon which $f$ is trained. She uses $\mathcal{D}_{aux}$ to train their her StyleGAN network. Alternatively, Eve can leverage a pretrained StyleGAN network available on the Internet, which removes the need for $\mathcal{D}_{aux}$. We also evaluate TMI on using $\mathcal{D}_{aux}$ with a significant deviation from the input distribution of $f$ (see §A.2.3). Lastly, Eve is a black-box adversary who cannot access the model parameters, gradients, or intermediate results while performing MI. Eve is only permitted to send a limited number of benign input queries to $f$ and use their output predictions. We emphasize that Eve is bound to a predefined query budget.

We note that Eve is even capable of populating an arbitrary prediction vector $\hat{y}$ or using only labels for conducting targeted MI. Under this scenario, Eve can apply label smoothing (Müller et al., 2019)

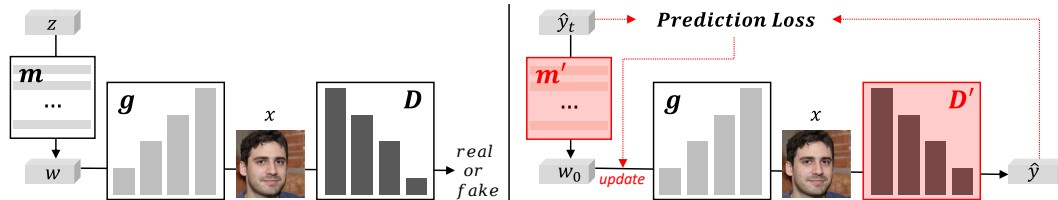

Figure 1: An overview of the TMI attack workflow (right), compared with the original StyleGAN network (left). The StyleGAN components modified for TMI ($m'$ and $f'$) are highlighted in red.

to hard-coded prediction outputs, each of which represents a corresponding class, then conduct the TMI attack. Please refer to §A.2.4 for further details.

## 4  DESIGN

The TMI attack consists of two distinct phases: *preparation* and *inversion*. Given query access to a target model $f$, the adversary leverages a pretrained StyleGAN network and alters its two components, the mapping network $m$ and the discriminator $D$, during the preparation phase. Once preparation is complete, any observed prediction output can be fed into the modified network to perform a offline MI attack to reconstruct its corresponding input image. The original and modified StyleGANs are illustrated in Figure 1.

### 4.1  PREPARATION PHASE: TWEAKING THE STYLEGAN NETWORK

**StyleGAN.** In a StyleGAN network, the generator is composed of two parts: a mapping network $m : \mathcal{Z} \mapsto \mathcal{W}$ and a synthesis network $g : \mathcal{W} \mapsto \mathcal{X}$. Unlike traditional GAN generators that take a random Gaussian vector $z$ from the latent space $\mathcal{Z}$ and pass it through the synthesis network, a mapping network $m$ first projects $z$ to an intermediate latent space $w \in \mathcal{W}$. The synthesis network $g$, consisting of consecutive style blocks, takes $w$ as the input for each style block. The style blocks synthesize images in a progressive manner, starting from low-resolution images and progressively refining them to higher resolutions. The discriminator $D$ in the StyleGAN network receives the final result from $g$ and determines whether this image is a real image or a synthesized sample. The minimax game between $D$ and $g$ gradually makes $D$ to better distinguish fake samples.

An important characteristic of the intermediate latent space $\mathcal{W}$ in StyleGAN is that image features are disentangled, meaning that different features are represented by separate dimensions in $\mathcal{W}$. This characteristic is encouraged during the StyleGAN training phase because (1) generating realistic images is easier when the representation is disentangled, which allows independent control over different image attributes, and (2) the separation of style blocks causes different subsets of $\mathcal{W}$ to contribute to different levels of styles, enabling fine-grained control over the generated images. This characteristic of disentangled image features in $\mathcal{W}$ has been found to be beneficial not only for style mixing (Karras et al., 2019) or editing (Abdal et al., 2019), but also as an effective basis for MI attacks. By manipulating specific dimensions in $\mathcal{W}$, an adversary is able to exert control over certain features of the generated image, enabling the reconstruction of input images corresponding to specific prediction vectors.

**Tailoring StyleGAN.** In TMI, the adversary leverages a publicly available StyleGAN network, or trains their own using $\mathcal{D}_{aux}$. In the evaluation section, we consider scenarios, including the use of pretrained StyleGAN networks, to assess the performance and effectiveness of the TMI attack.

The original mapping network $m$ is trained to convert a Gaussian vector $z$ into an intermediate latent $w$. Therefore, the adversary trains a new mapping network $m' : \hat{\mathcal{Y}} \mapsto \mathcal{W}$ to emit $w$ directly from an observed prediction vector $\hat{y}$. For this, the adversary exploits the StyleGAN network to generate triplets, each of which consists of an image $x$ generated via the generator using a random vector $z$, the intermediate latent $w$ used to generate $x$, and a prediction vector $\hat{y}$ obtained from $f$ by querying $x$. That is, the adversary generates $\mathcal{D}_{gen} = \{(w, x, \hat{y}) \mid z \in \mathcal{Z},\ w = m(z),\ x = g(w),\ \hat{y} = f(x)\}$. After the dataset is complete, $m'$ is trained to minimize the following loss function:

$$\mathcal{L}_{m'} = \mathbb{E}\left[(w - \hat{w})^2\right], \text{ where } \hat{w} = m'(\hat{y}). \tag{2}$$

Although this training procedure does not require white-box access to $f$, it still requires sending a number of queries (i.e., $|\mathcal{D}_{gen}|$) to $f$ during $\mathcal{D}_{gen}$ construction. We set this number to be 100k throughout our main evaluation, which is significantly smaller than the number of required queries in prior works summarized in Table 1.

The new mapping network $m'$ plays a key role in locating an initial latent point for each observed prediction $\hat{y}$. We expect $m'$ to learn a way of distilling a style given a prediction vector during its training procedure, which the adversary exploits in the later inversion phase (§4.2). We exemplify the efficacy of $m'$ in selecting a reliable initial latent point $w_0$ with high fidelity and the superiority of this approach compared to the prior MI methods.

Lastly, the adversary conducts transfer learning, to make a surrogate model $f' : \mathcal{X} \mapsto \hat{\mathcal{Y}}$ from the original discriminator $D$ in the StyleGAN network with its last layer changed to match the number of classes in $f$. The training of $f'$ minimizes the following loss:

$$\mathcal{L}_{f'} = \mathbb{E}\left[(\hat{y} - f'(x))^2\right]. \tag{3}$$

The goal of $f'$ is to emit a prediction vector similar to the one produced by $f$ for each $x \in \mathcal{D}_{gen}$. This process does not send additional queries to $f$ as it only leverages $\mathcal{D}_{gen}$ which is already obtained from the previous step of training $m'$.

## 4.2 INVERSION PHASE: RECONSTRUCTING INPUT IMAGES

**Algorithm 1** TMI Attack

1: **procedure** ATTACK(target $\hat{y}_t$, iteration $n$, step size $\eta$, exploration $e$, mix probability $\delta$)
2:     $w_0 \leftarrow m'(\hat{y}_t)$
3:     $\ell_{best} \leftarrow \infty$
4:     **for** $i \in \{0, \ldots, n\}$ **do**
5:         $x_i \leftarrow g(w_i)$
6:         $y'_i \leftarrow f'(x_i)$
7:         $\ell_i \leftarrow Distance(y'_i, \hat{y}_t)$    ▷ $\ell_2$ Loss
8:         **if** $\ell_i < \ell_{best}$ **then**
9:             $x_{best}, \ell_{best} \leftarrow x_i, \ell_i$
10:        **end if**
11:        $w_{i+1} \leftarrow w_i - \eta\nabla_w \ell_i$
12:        **if** $i$ is multiple of $e$ **then**
13:            $w_{i+1} \leftarrow RandomMix(w_{i+1}, \delta)$
14:        **end if**
15:     **end for**
16:     **return** $x_{best}$
17: **end procedure**

Once the preparation phase is complete, the adversary can launch the inversion phase on any observed target prediction $\hat{y}_t$ to reconstruct its input image. In TMI, white-box optimization using $f$'s gradients is replaced with repetitive approximated optimization, starting from $w_0$ derived from the renewed mapping network $m'$:

$$w_{t+1} := w_t - \eta\nabla_w \left[f'(g(w_t)) - \hat{y}_t\right]^2. \tag{4}$$

Algorithm 1 describes the overall process of the inversion phase. The adversary starts by obtaining an initial latent representation $w_0 = m'(\hat{y}_t)$ using $m'$ obtained from the previous phase (Line 2). This initial latent is fed into $g$ to generate an image $x$ (Line 5). In Lines 6–7, this synthesized image is fed into $f'$ to produce a prediction result $y'$, and it then computes the distance between $y'$ and $\hat{y}$. In Line 11, $w$ is optimized via gradient descent so that the generated image produces an approximated prediction $y'$ that is closer to $\hat{y}$.

For every $e$ steps, TMI performs $RandomMix$, where subsets of $w$ are reset to $w_0$ with the probability $\delta$. This is done to avoid $w$ from overfitting only to a specific style, which leads to unnatural images. $RandomMix$ prevents the optimization routine from getting trapped in a local minimum and allows it to explore different styles and combinations. Finally, the algorithm returns $x$ that recorded the closest $y'$ to $\hat{y}_t$.

Note that in TMI, the adversary exploits $f'$ in an offline manner to refine the initial latent vector $w_0$, which is also obtained by an offline single-pass to $m'$. Therefore, TMI does not generate any queries or require white-box access to $f$ throughout the inversion phase. This makes the attack completely passive once the preparation phase is complete.

## 4.3 SUMMARY AND DIFFERENCES TO PRIOR MI ATTACKS

[1]We use 100k as default for evaluations. We also tested different query budgets that start from 5k.

[2]LO-MI does not have an explicit query limit on `Prep`, however we observed it to have the highest number of queries in practice. Hence, we regard it as the upper-bound among baselines.

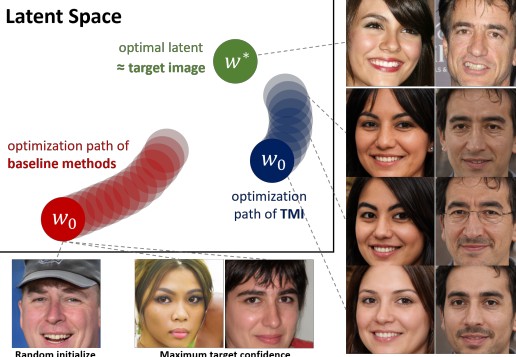

Figure 2: Latent space exploration of TMI versus baseline methods.

Table 1: Comparison of the required queries. The number of queries under the `Attack` is required for each attack attempt, whereas numbers under `Prep` is required once in the preparation phase.

| Method | Query Count | | Image |
|---|---|---|---|
| | Prep | Attack | Prior |
| AMI | $|\mathcal{D}_{aux}|$ | 0 | None |
| TMI (ours) | $|\mathcal{D}_{gen}|$[1] | 0 | |
| MIRROR-w | 100k | 160k | Style- |
| P&P | 5k | 34k | GAN |
| MIRROR-b | 100k | 10k | |
| RLB-MI | 0 | 80k | GAN |
| LO-MI | 100k[2] | 2k | |

Previous studies have focused on reconstructing class-generic images, overlooking the reconstruction of instance-specific features. This trend comes from the fact that GAN- and StyleGAN-based MI performs optimization from either a random initial latent (Kahla et al., 2022; Han et al., 2023) or the latent that yields the maximum target confidence (An et al., 2022; Struppek et al., 2022; Yuan et al., 2023). Such initial points tend to be biased toward singularities and become far from the optimal point as shown in Figure 2. Prior works overcome this issue by performing a large number of optimization steps, which naturally demand an excessive number of input queries, thus undermining the practicality of MI. Table 1 shows the number of queries for each MI approach. Furthermore, using a large number of optimization steps frequently leads to convergence to local optima due to the nature of greedy updates. In contrast, we employ the customized mapping network $m'$ to directly project the predictions to the latent space. By pinpointing a reliable starting point via $m'$, TMI bypasses most of the early optimizations present in existing MI attack methods.

The choice of constructing $f'$ also brings benefits compared with a naive black-box migration of white-box methods, which would be to train a surrogate model from scratch as a substitute for $f$ in the optimization routine. Transfer learning from $D$ offers a more reliable and generalized surrogate model due to the fact that $D$ has already been exposed to numerous styles of images during the training of StyleGAN. We show that a surrogate model trained from scratch is insufficient to imitate the optimization path of the target model $f$, in our ablation study (see §A.2.1).

## 5 EXPERIMENTS

We conducted a comprehensive comparison of the inversion capability of TMI with state-of-the-art MI attacks, including both black-box and white-box methods. The black-box methods include MIRROR-b, RLB-MI (Han et al., 2023), LB-MI (Kahla et al., 2022), and AMI (Yang et al., 2019), while the white-box methods include MIRROR-W and P&P (Struppek et al., 2022). MIRROR-b and MIRROR-w represent the black-box and white-box MI methods using MIRROR (An et al., 2022), respectively.

### 5.1 EXPERIMENTAL SETUP

We selected two tasks for MI: facial recognition and chest X-ray diagnosis. For facial recognition, we prepared target networks trained on the FaceScrub (Ng & Winkler, 2014) and CelebA (Liu et al., 2015) datasets. We used ResNeSt-101 (Zhang et al., 2020a) as the architecture for $f$ in our main experiments. Refer to § A.2.2 for experiment results on other architectures. As for the adversary's StyleGAN network, we used a publicly available StyleGAN2 (Karras et al., 2020) network trained on the Flickr-Faces-HQ (FFHQ) (Karras et al., 2019) dataset. We note that the underlying distribution of the StyleGAN network is different from that of the target networks, reflecting a practical attack setting. We have verified that TMI remains effective under conditions of an even greater distribution shift; these findings are detailed in § A.2.3. For chest X-ray diagnosis, we used a StyleGAN2 network trained on the NIH Chest X-ray dataset (Wang et al., 2017). The target network was trained using the PadChest (Bustos et al., 2019) dataset. Similar to the facial recognition task, datasets used for training the two parties (adversary and target network) have different distributions. We employed same StyleGAN2 networks as image priors in all baseline methods using StyleGAN. For other attacks

that incorporate GANs (Kahla et al., 2022; Han et al., 2023), we trained their GANs using the same dataset used to train StyleGANs. For more details on the datasets and models, please refer to § A.1.

## 5.2 EVALUATION METRICS

To evaluate the effectiveness of TMI, we used several metrics to assess the quality of the inversion results and the instance-specific MI capabilities. Accuracy and feature distance are commonly used metrics in the MI literature (An et al., 2022; Struppek et al., 2022; Wang et al., 2021; Zhang et al., 2020b). We also used two additional metrics, class-wise coverage and attribute accuracy, to demonstrate the capabilities of instance-specific MI.

**Accuracy (`Acc@1` and `Acc@5`).** To assess the resemblance of the reconstructed images to the target image class, we computed the proportion of reconstructed images that were classified into the same class as the target image by $f_E$, an evaluation classifier using Inception-v3 (Szegedy et al., 2016) trained on the same dataset as $f$. This proportion represents the accuracy of the reconstruction process, indicating how well the reconstructed images capture the features of the target class in general.

**Feature distance (`F-dist`).** The feature distance metric captures the similarity between two images at an intermediate representation layer (Dosovitskiy & Brox, 2016) of $f_E$, which quantifies the perceptual similarity between the images. Specifically, we computed the average $\ell_2$ distance between features extracted from the penultimate layer of a $f_E$, hence computing the similarity in high-level visual features perceived by the classifier (Zhao et al., 2021).

**Class-wise coverage (`Cover`).** We adopted the class-wise coverage metric to assess whether the reconstructed samples successfully captured the intra-class diversity, which is crucial in the instance-specific MI task. We use a slightly modified version of the original notion introduced by Naeem et al. (2020). This metric evaluates the extent to which the reconstructed samples cover the range of variations within each target class. It measures the fraction of target images that have a reconstructed sample in close proximity, providing insight into how well the reconstruction process captures the intra-class diversity. The class-wise coverage is formally defined as follows:

$$\texttt{Cover} = \frac{1}{N} \sum_{i=1}^{N} 1_{\exists\, j \text{ s.t. } Y_j \in B(X_i,\, NND_k(X_i))}. \tag{5}$$

where $N$ and $1_{(\cdot)}$ are the number of samples and the indicator function, respectively. Whereas the original notation considered the intermediate representations of real and fake samples as $X_i$ and $Y_i$, we replaced them with the intermediate representations of the target and reconstructed images, respectively. $B(x, r)$ indicates a sphere in the representation space around $x$ with radius $r$, and $NND_k(X_i)$ denotes the distance from $X_i$ to its $k^{\text{th}}$ nearest neighbor. We used $k = 1$ throughout our evaluations.

**Attribute accuracy.** To evaluate the success of feature reconstruction, we trained attribute classifiers using the Inception-v3 architecture (Szegedy et al., 2016), where the final layer of the classifier was adjusted to accommodate the number of categories for each attribute. Specifically, we trained attribute classifiers using the respective attribute labels available in CelebA and the gender and age information in PadChest.

## 5.3 EXPERIMENTAL RESULTS

Figure 3 shows the inversion results obtained by other MI methods and TMI. It is evident that the samples reconstructed using TMI are visually more similar to their corresponding original images, making it easier to identify them as the same identity. We also note that TMI reconstructs facial expressions (columns 1, 2, 4), and instance-specific attributes (columns 3, 5) such as glasses. The other methods did succeed in reconstructing some general features of the original images. However, they failed to capture the fine and specific characteristics of the original image. Moreover, we observed that RLB-MI, LO-MI, and AMI methods are not suitable for real-world MI attacks on high-dimensional images. While the authors had demonstrated their success on $\mathbb{R}^{3\times64^2}$ tightly-cropped images, we found that these black-box attacks experienced difficulties in reconstructing $\mathbb{R}^{3\times224^2}$ input images. In contrast, the white-box attacks exhibited high accuracy since they explicitly took into account the classification loss on the target model during their optimization steps.

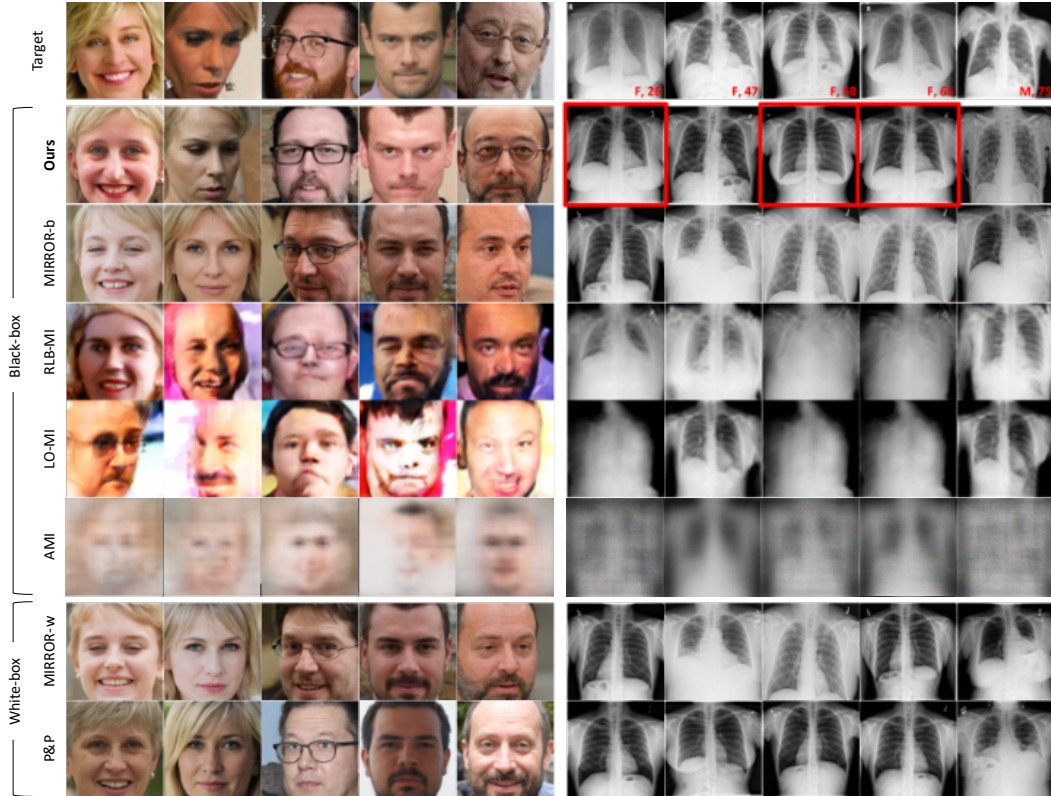

Figure 3: Comparison of the inversion results on facial recognition (left) and chest X-ray diagnosis (right). Gender and age information for each X-ray target images are shown in red.

Table 2: Inversion performance of TMI and SOTA methods. The top five MI methods are black-box attacks, and the remaining ones are white-box attacks. The best performing black-box attack metrics are marked in bold.

| | Method | Acc@1 ↑ | Acc@5 ↑ | F-dist ↓ | Cover ↑ | | Method | Acc@1 ↑ | Acc@5 ↑ | F-dist ↓ | Cover ↑ |
|---|---|---|---|---|---|---|---|---|---|---|---|
| **Facial recognition** | TMI (ours) | $\mathbf{.3408}_{\pm.0061}$ | $\mathbf{.6255}_{\pm.0092}$ | $\mathbf{.2950}_{\pm.0009}$ | $\mathbf{.2067}_{\pm.0126}$ | **Chest X-ray diagnosis** | TMI (ours) | $.5158_{\pm.0127}$ | $\mathbf{.9999}_{\pm.0004}$ | $\mathbf{.0851}_{\pm.0006}$ | $\mathbf{.2415}_{\pm.0396}$ |
| | AMI | $.0443_{\pm.0179}$ | $.0906_{\pm.0278}$ | $.3860_{\pm.0011}$ | $.0033_{\pm.0009}$ | | AMI | $.0743_{\pm.0080}$ | $.8717_{\pm.0103}$ | $.1788_{\pm.0008}$ | $.0002_{\pm.0002}$ |
| | MIRROR-b | $.2026_{\pm.0267}$ | $.4533_{\pm.0394}$ | $.3564_{\pm.0058}$ | $.0613_{\pm.0059}$ | | MIRROR-b | $\mathbf{.7786}_{\pm.0757}$ | $.9983_{\pm.0049}$ | $.1094_{\pm.0183}$ | $.0172_{\pm.0154}$ |
| | RLB-MI | $.2568_{\pm.0172}$ | $.5044_{\pm0246}$ | $.3804_{\pm.0049}$ | $.0514_{\pm.0038}$ | | RLB-MI | $.0790_{\pm.0000}$ | $.8988_{\pm.1026}$ | $.1827_{\pm.0069}$ | $.0002_{\pm.0000}$ |
| | LO-MI | $.2611_{\pm.0079}$ | $.5155_{\pm.0115}$ | $.3921_{\pm.0008}$ | $.0536_{\pm.0038}$ | | LO-MI | $.0790_{\pm.0000}$ | $.9860_{\pm.0000}$ | $.1769_{\pm.0006}$ | $.0002_{\pm.0000}$ |
| | P&P | $.7779_{\pm.0308}$ | $.9476_{\pm.0076}$ | $.2470_{\pm.0029}$ | $.1440_{\pm.0059}$ | | P&P | $.7250_{\pm.2769}$ | $.9960_{\pm.0057}$ | $.1155_{\pm.0211}$ | $.0084_{\pm.0074}$ |
| | MIRROR-w | $.8129_{\pm.0228}$ | $.9531_{\pm.0085}$ | $.2491_{\pm.0038}$ | $.1257_{\pm.0048}$ | | MIRROR-w | $.8634_{\pm.1339}$ | $.9983_{\pm.0049}$ | $.1138_{\pm.0264}$ | $.0137_{\pm.0140}$ |

Table 2 provides the quantitative evaluation results of TMI, along with state-of-the-art MI methods. The experimental results clearly demonstrate that TMI outperformed all other black-box MI attacks according to the reported metrics. These results confirm the superiority of TMI in conducting practical black-box MI.

We emphasize the significant decrease in the number of required queries in performing MI. When assuming a scenario in which the adversary aims to generate 530 class facial images in the FaceScrub dataset, TMI requires the default query budget of only 100k queries to a target model in achieving the reported metrics in Table 2. In contrast, MIRROR-w and MIRROR-b, which performed the best beside TMI, required $84900k = 100k + 160k \times 530$ and $5400k = 100k + 10k \times 530$ queries, respectively. Outside of AMI which failed to produce any meaningful results, LO-MI required the least number of queries among the baselines: $1160k = 100k + 2k \times 530$, which is still significantly higher than the one that TMI required.

In addition, we conducted experiments to investigate the change in reconstruction performance while varying the size of $D_{gen}$ that is used during the preparation phase to train $m'$ and $f'$. In Figure 4, we compared the MI performance of TMI with different query budgets: 5k, 10k, 50k, and 100k (default).

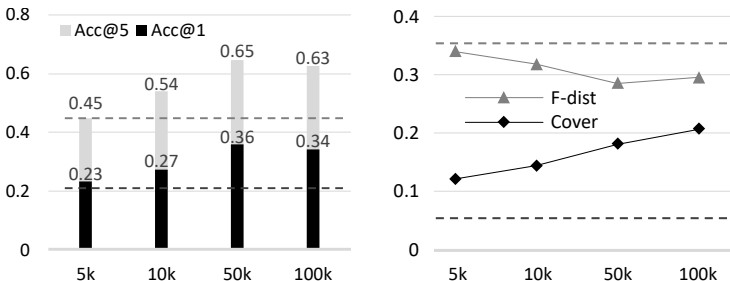

Figure 4: Changes in the MI performance while varying the query budget. Dotted lines indicate the performance of MIRROR-b.

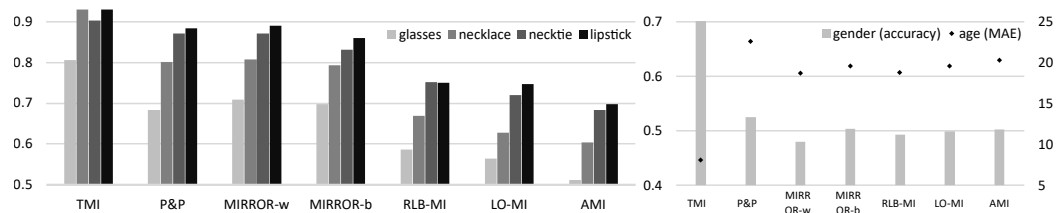

Figure 5: Attribute accuracy on facial recognition (left) and chest X-ray diagnosis (right). For age, we reported the mean absolute error.

Observe that with only 5k queries, TMI attained similar performance to the one of MIRROR-b that required a total of 5400K queries, resulting in a 1,080 times decrease in the query budget.

For chest X-ray diagnosis, TMI significantly outperformed all other methods by a large margin. The performance gains can be attributed to the unique characteristics of the chest X-ray classifier. Unlike facial recognition, where each class corresponds to a single identity, the classes in the chest X-ray classifier encompass a diverse range of identities and features. For example, the *pneumonia* class includes samples from both male and female individuals, and the age of the subjects spans across various age groups. In this particular context, we argue that traditional metrics such as Acc@1 and Acc@5 are not indicators of instance-specific privacy leakage. Thus, we additionally used intra-class metrics (i.e., F-dist, Cover, and attribute accuracy) to evaluate TMI.

Figure 3 presents a comparison of the inversion results on chest X-ray images. Notably, TMI successfully captures private information such as gender and body shape. For example, the highlighted TMI inversion results in the second row clearly shows a female X-ray image, even to the human eye, due to the accurate reconstructions of the chest shape when comparing its reconstruction quality with the other baseline results.

To emphasize the capability of targeted MI, we further conducted comparison evaluations that measure the attribute accuracy across instance-specific attributes in Figure 5. As the figure shows, TMI strictly outperformed all other methods in capturing subtle and intra-class features. For example, the attribute classifier for checking glasses on the TMI reconstructed facial images reported an accuracy of 80.7% while MIRROR-b and P&P reported 69.8% and 68.3%, respectively. Also, when inferring the ages of the reconstructed chest X-ray images, TMI-generated images contribute to reporting a mean absolute error (MAE) of 8.1198, significantly outperforming all other methods. These results highlight the capability of TMI to capture instance-specific private information compared to the baselines.

## 6 CONCLUSION

We have proposed TMI, a novel black-box MI attack that achieves instance-specific MI using a limited query budget. TMI alters the mapping network of a benign StyleGAN network to find a reliable initial latent point corresponding to a target prediction output, then performs further optimization by leveraging a surrogate model distilled from the StyleGAN discriminator. TMI significantly decreases the number of required queries while improving the reconstruction quality over state-of-the-art black-box MI methods.

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

# A  APPENDIX

## A.1  EXPERIMENTAL DETAILS

All experiments took place on an Ubuntu 20.04.1 system with CUDA 11.3, on top of 512GBs of RAM, two Intel Xeon Gold 6258R CPUs, and four RTX 3090 GPUs. We used Python 3.10.11 with PyTorch 1.12.1 and Torchvision 0.13.1. Please refer to our source code for further information regarding auxiliary Python packages and versions used throughout the experiments.

### A.1.1  DATASETS AND MODELS

We used the official bounding-box information for FaceScrub and use its 530 identities as each class for classification. For CelebA, we randomly selected 1000 identities from the entire CelebA dataset for classification. We applied 108×108 center-crop to the aligned version of CelebA images to align them with the FaceScrub images. For Chest X-ray, we used only the front-facing X-ray images, and selected the seven most frequent findings (normal, pneumonia, tuberculosis sequelae, emphysema, heart insufficiency, pulmonary fibrosis, COPD signs) from the PadChest dataset for disease classification. Throughout the experiments, the input to the target models were unified to 224×224, and 299×299 for the evaluation classifiers. Input images were resized to match the respective input dimensions. We used bilinear interpolation for every resizing operation. StyleGAN generated images given as input to $f$ during the preparation phase should also be cropped & resized appropriately. We applied 180×180 crop, then resized them to match the input image size (224×224 or 299×299). We used pretrained ImageNet checkpoints provided by Torchvision or PyTorch Hub as initial weights for the target and evaluation classifiers, and replaced their final fully-connected layer to match the number of classes in respective datasets. 10% of each dataset were used as the test split.

As for the StyleGAN image prior, we used a pretrained StyleGAN2 available online [3] to attack the facial recognition domain. For targeting the chest X-ray diagnosis task, we used a custom StyleGAN2 network trained upon the NIH Chest X-ray dataset. For simplicity, we unified the StyleGAN networks to generate 256×256 images. We kept the GAN architecture of the original implementation for baselines RLB-MI and LO-MI. For each baseline, we used GANs custom-trained on FFHQ and NIH Chest X-ray images, respectively. Pretrained networks uploaded by their authors were unusable since they used a much tighter crop compared to the FaceScrub official bounding-box. We also observed worse results when the image priors were replaced with StyleGAN.

Note that the TMI adversary has no knowledge of the cropping or resizing logic of $f$, as it is processed inside the black-box service. Accordingly, $f' : \mathcal{X} \mapsto \hat{\mathcal{Y}}$ receives the full images generated from StyleGAN, whereas the actual $\hat{\mathcal{Y}}$ is calculated inside $f$ upon cropped & resized versions.

### A.1.2  TMI ATTACK

In constructing $D_{gen}$, we applied the truncation trick, which is a generally used technique to promote natural image generation with StyleGAN (Karras et al., 2019). Specifically, we generated images with truncation $\psi = 0.7$ in order to avoid unnatural synthesis results. To increase the dispersion between confidence values for easier training of $m'$ and $f'$, we apply natural logarithm to the prediction vectors from $f$.

Throughout the inversion phase of the TMI experiments, we used $n = 5000$, $\eta = 10^{-5}$, $e = 500$, $\delta = 0.05$ for Algorithm 1. Note that despite these hyperparameters can be fine-tuned to each attack scenario, we fixed them for simplicity and to demonstrate the robustness of TMI.

In addition, to bound the reconstructed images to the natural image domain, we applied a clipping technique right after $RandomMix$ in Line 13 of Algorithm 1. Formally, we computed the dimension-wise mean $\mu$ and deviation $\sigma$ of $\mathcal{W}$ from $D_{gen}$, then:

$$w^{(i)} = \max(\ \min(w^{(i)}, \ \mu^{(i)} + \sigma^{(i)}), \ \mu^{(i)} - \sigma^{(i)}), \tag{6}$$

where $\cdot^{(i)}$ denotes the $i^{\text{th}}$ dimension. Note that removing the clipping logic results in similar metric scores, however the reconstructed images often appears visually unnatural.

---

[3] https://github.com/rosinality/stylegan2-pytorch

For each evaluation scenario, we attacked 1000 randomly selected samples from the test split of $f$'s dataset. This simulates the real-world attack, where the target images correspond to one of $f$'s classes, however not directly included in its training set. For TMI and AMI in Table 2, we repeated the experiment 8 times and reported the mean and standard deviation of each metrics. For the other baseline attacks, we selected the 8 best candidates from their final output and report their mean and standard deviation.

## A.2 ADDITIONAL EXPERIMENTS

In this section, we demonstrate results of experiments outside of the main experiment in § 5 in order to provide deeper understanding on the behaviour of TMI under different situations.

### A.2.1 ABLATION STUDY

Table 3: MI performance comparison between initialization methods. Best cases are marked in bold.

| Method | Acc@1 | Acc@5 | F-dist | Cover |
|---|---|---|---|---|
| $m'$ only | .0840 | **.2620** | **.1523** | **.0817** |
| Maximum target confidence | **.0900** | .2200 | .4135 | 0 |
| Random initialize | .0010 | .0100 | .4941 | .0309 |

**Effect of the New Mapping Network.** In order to assess the efficacy of the new mapping network $m'$ in locating a reliable initial latent point ($w_0$), we evaluated the synthesis result directly from $w_0$ (i.e., $g(w_0)$) without the optimization steps, and compared it with initialization techniques of existing methods. The results in Table 3 suggest that $w_0$ is closer to the target image in terms of F-dist and Cover compared to random initialization or maximum target confidence.

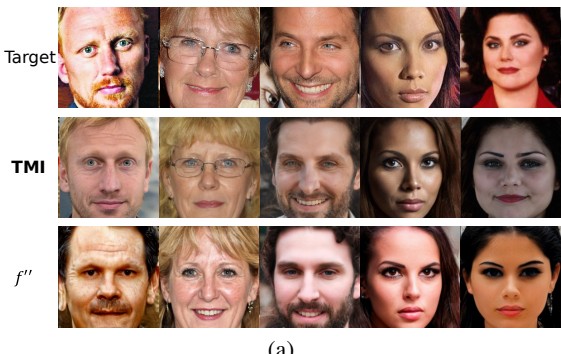

|  | Performance degradation |
|---|---|
| Acc@1 ↑ | 0.3408→0.1100 (67.72% ↓) |
| Acc@5 ↑ | 0.6255→0.2660 (57.47% ↓) |
| F-dist ↓ | 0.2950→0.3816 (29.35% ↑) |
| Cov ↑ | 0.2067→0.0982 (52.49% ↓) |

(a)                                                          (b)

Figure 6: Qualitative (a) and quantitative (b) comparison of TMI and $f''$. In (b), values under parenthesis indicate the percentage of degradation from TMI to $f''$.

**Effect of the Surrogate Model.** A naive black-box migration of the white-box approaches would be to train a surrogate model from scratch, instead of using $f'$ (which is transfer-learned from $D$ of the StyleGAN network). Similar to $f'$, this new surrogate model works as a substitute for $f$, removing the white-box requirement. We refer to such MI attack scenario as $f''$. Here, we demonstrate that $f''$ is insufficient, thus justifying the use of $f'$. We empirically found that $f''$ is insufficient to provide reliable optimization. As shown in Figure 6, the reconstructions from $f''$ only succeeds in capturing some general coarse-grained features, failing to reconstruct the instance-specific details or even the identity. We argue that it is necessary to use $f'$ and take advantage of its pre-exposure to various styles during the StyleGAN training.

Table 4: MI performance across scenarios involving different target domains and $f$.[4]

| Scenario | Arch | Acc@1 | Acc@5 | F-dist | Cover |
|---|---|---|---|---|---|
| FFHQ ↓ FaceScrub | MobileNet-v3 | .3380 | .6390 | .2878 | .1813 |
| | ResNeSt-101 | .3408 | .6255 | .2950 | .2067 |
| | DenseNet-169 | .3265 | .6035 | .3027 | .1967 |
| FFHQ ↓ CelebA | MobileNet-v3 | .2080 | .4345 | .9215 | .2343 |
| | ResNeSt-101 | .2880 | .5520 | .8503 | .2738 |
| | DenseNet-169 | .2015 | .4335 | .5751 | .2965 |
| CXR14 ↓ PadChest | MobileNet-v3 | .3540 | .9880 | .1161 | .2316 |
| | ResNeSt-101 | .5158 | .9999 | .0851 | .2415 |
| | DenseNet-169 | .4840 | .9940 | .1188 | .1972 |

### A.2.2   DIFFERENT TARGET MODEL ARCHITECTURES.

In addition to ResNeSt-101, we further investigated how MI performance varies across different architectures for $f$. Specifically, we investigated DenseNet-169 (Huang et al., 2017) and MobileNet-v3 (Howard et al., 2019) across different target domains and observed that the tendency of the results were constant to the main evaluation throughout all configurations (Table 4). For example, in the FFHQ→FaceScrub scenario, TMI surpassed all black-box baselines in every metric. This suggests that TMI is generally applicable across various target systems under a truly black-box setting, i.e., in a target model-agnostic manner.

### A.2.3   $\mathcal{D}_{aux}$ WITH SIGNIFICANT DEVIATION FROM $\mathcal{D}$

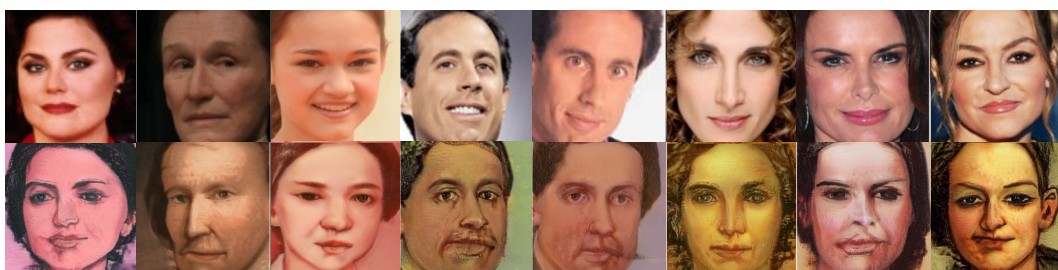

Figure 7: Target FaceScrub images (top) and TMI-reconstructed using StyleGAN trained on art-style face portraits (bottom).

Sometimes the TMI adversary may not be able to obtain $\mathcal{D}_{aux}$ with a distribution similar to $\mathcal{D}$. In this experiment, we show that TMI is robust enough to capture features of the target images $x \in \mathcal{D}$ that are embedded in the distribution of $\mathcal{D}_{aux}$, even when $\mathcal{D}$ and $\mathcal{D}_{aux}$ as a whole have distinct distributions. We used a StyleGAN trained on art portraits [5] as the image prior to attack a ResNeSt-101 network trained to classify FaceScrub identities. From Figure 7 clearly demonstrates that TMI can successfully reconstruct input images with high fidelity, including facial features, posture, and rough color.

### A.2.4   LABEL-ONLY TMI

Real-world MLaaS services often returns only the predicted label instead of the full confidence vector. A slight modification to the TMI workflow extends the attack to support these situations. Specifically, we implement label smoothing on each prediction output received from the target model, converting each label prediction into a confidence vector. Given the confidence vector, TMI can operate in the same way as the original version. Table 5 demonstrates that TMI is still effective in label-only

---

[4]Under Scenario, A→B indicates that the attacker utilizes an image prior (i.e., StyleGAN) obtained from A to attack $f$ that was trained to classify B images.

[5]https://github.com/ak9250/stylegan-art

Table 5: MI performance on label-only setting. The original TMI and MIRROR-b experient results are also included for comparison.

| Method | Acc@1 | Acc@5 | F-dist | Cover |
|---|---|---|---|---|
| TMI (original) | .3804 | .6255 | .2950 | .2067 |
| Label-only TMI | .2399 | .4800 | .3637 | .1167 |
| MIRROR-b | .2026 | .4533 | .3564 | .0613 |

situations. While there is a slight drop in performance compared to the original use-case, notice that the attack continues to surpass the capabilities of MIRROR-b in all metrics outside of F-dist.

