# OpenReview forum: "Targeted Model Inversion: Distilling Style Encoded in Predictions"
_ICLR.cc/2024/Conference — Submitted to ICLR 2024_

### Official Review · Reviewer_wJqb · 2023-10-27

**Soundness:** 3 good
**Presentation:** 2 fair
**Contribution:** 2 fair
**Rating:** 6
**Confidence:** 3

**Summary:**

The paper introduces a new Model Inversion (MI) method that exploits the hierarchical latent features of a separately trained StyleGAN architecture. A goal of MI is to recover the input x of a classifier given the output probability vector y. The MI model can then be used for malicious attacks aimed at extracting private information, since the predictions y are easier to obtain than x in most privacy regulations. The new MI methods show substantially higher performance than a wide range of both black-box and white box baselines on several image datasets.

**Strengths:**

- The approach is simple but well-motivated.
- The recovered images are substantially better than the baseline models both qualitatively and quantitatively.
-The paper is well-written and the methodology is well-explained.
-The experiments are rigorous and rather comprehensive.

**Weaknesses:**

I am struggling to find a good societal application of this work as the explicit aim of the paper is to improve performance of a family of malicious attacks that can be used to leak private information. While I do agree that open research on attacks is important, it seems to me that this work is very helpful to potential attackers while not providing any real insight concerning possible defense strategies.  Note that the paper does not introduce a conceptually novel way to perform attacks, which would provide important information to the public. Instead, it offers a highly optimized approach that exploits several, rather standard, techniques. For this reason, I am not convinced that a paper like this has a place in a machine learning conference.

I do appreciate the technical skills shown by the authors, I think that equal effort should be spent in considering the societal implications and in discussing possible defense strategies.

Apart from ethical consideration, I find the domain of application to be rather narrow and more suitable for a more specialized venue. All in all, the paper does not introduce any conceptually new technique since the use of proxy classifiers and generative models is common in the reconstruction literature.

Update:
Given the author's revision, I decided to increase my score to 6 and to recommend acceptance.

**Questions:**

Could you discuss the ethical implications of your work and provide some insights concerning possible defense methods?

**Details Of Ethics Concerns:**

The paper reads too much as a manual on how to perform malicious attacks. I do see the value of attack research but I think that this might cross the line.

Update:
I believe that the authors responded appropriately to my concerns, which are addressed in the revised manuscript

---

> ### Author Response · Authors · 2023-11-21
> **Author Response to Reviewer wJqb (1/2)**
>
> We appreciate your helpful feedback. We address each concern in the following:
>
> **Narrow domain of application; nothing conceptually new**
>
> Please note that we demonstrated that retraining a mapping network in StyleGAN contributes to better distilling styles in prediction vectors, which provides promising initial datapoints for MI in the inversion phase. We also extensively evaluated the efficacy of this new MI attack, demonstrating the feasibility of distilling style in prediction for better MI.
>
> To demonstrate the wide range of TMI’s applicability, we conducted two additional experiments: (1) using another generative model as prior, and (2) adopting TMI to a white-box attack scenario.
>
> For (1), we replaced StyleGAN with UNet-GAN [6], and compared the results to the base experiment in the table below. For the surrogate model $f’$, we used the left-half of the UNet-GAN’s discriminator. Since UNet-GAN does not incorporate a mapping network, a custom mapping layer was trained from scratch. Observe that while the overall performance is slightly decreased compared to TMI with StyleGAN, it is still more effective than MIRROR-b. The slight drop in performance is largely due to the entangled latent space of UNet-GAN, where the latent space $\\mathcal{Z}$ is directly used without mapping it to an intermediate disentangled latent space $\\mathcal{W}$ in advance. We conclude that while StyleGAN is still the most effective image prior to be utilized, other GAN-based methods can generally benefit from our suggested approach of leveraging a new mapping layer and distilling the discriminator for a surrogate model. Therefore, the contribution of TMI is not only limited to the packaged attack method as a whole, but also the individual components that can be applied independently.
>
> | | **Acc@1 ↑** | **Acc@5 ↑** | **F-dist↓** | **Cover ↑** |
> |-|:-:|:-:|:-:|:-:|
> | TMI | .3804 | .6255 | .2950 | .2067 |
> | TMI with UNet-GAN | .2201 | .4811 | .3254 | .2063 |
> | MIRROR-b | .2026 | .4533 | .3564 | .0613 |
>
> For (2), we used the original model instead of the surrogate model, and the performance gains are listed in the table below. As expected, the performance is improved by an average of 63.7%, outperforming the white-box attacks across all four metrics. This also indicates that retraining a mapping network can be applied for conducting  white-box TMI attacks to improve their reconstruction performance. We will include this result in A.2.1. Effect of the Surrogate Model.
>
> | | **Performance gain** |
> |:-:|:-:|
> | Acc@1 ↑ | 0.3408 → 0.8264 |
> | Acc@5 ↑ | 0.6255 → 0.9534 |
> | F-dist ↓ | 0.2950 → 0.1975 |
> | Cover ↑ | 0.2067 → 0.2619 |
>
> Our additional experiment results demonstrate that the distillation process of TMI is a generally-applicable technique which can not only support using other generative models as prior, but also augment existing white-box attacks.
>
> Moreover, prior MI attacks have primarily focused on reconstructing class-generic features, which harms privacy of individuals that directly contribute to the training phase of the target model. Individuals who interact with the target model only after deployment (i.e., their user data are not included in the training set) were not exposed to privacy leakage. However,  our experimental results suggest otherwise. Since TMI is capable of reconstructing the original image specific to an output prediction, it demonstrates the privacy threat of reconstructing inputs of ML service users. These results also shed light on information leakage from output predictions.

---

> ### Author Response · Authors · 2023-11-21
> **Author Response to Reviewer wJqb (2/2)**
>
> **Ethical implications and possible defense methods**
>
> We believe that investigating a new attack and concretizing its performance and impact is an important contribution that motivates further research in mitigating the addressed threat. Numerous previous studies regarding adversarial attacks (e.g. FGSM [1], C&W [2]) and MI attacks (e.g. MIRROR, P&P) proposed novel and powerful attacks demonstrating new upper bounds of robustness and resiliency of ML models against the proposed attacks.
>
> Additionationally, we will discuss possible mitigation methods in the revised paper. Specifically, we will discuss the downgraded MI performance when the target model only returns a label for a given query (see A.2.4) and another mitigation of injecting random noise to a prediction output (while preserving its prediction label). Observe that its performance dropped below MIRROR-b, in particular, F-dist and Cover metrics are greatly degraded from the original TMI experiment (21.1% and 81.6%). This suggests that the noise successfully distracts TMI from reconstructing subtle features. The table below shows the performance of TMI against target models that inject random noise to prediction outputs. Finally, we will explain the applicability of recent MI defenses [3-5].
>
> | **Method** | **Acc@1 ↑** | **Acc@5 ↑** | **F-dist ↓** | **Cover ↑** |
> |-|:-:|:-:|:-:|:-:|
> | TMI (original) | .3804 | .6255 | .2950 | .2067 |
> | Label-only TMI | .2399 | .4800 | .3637 | .1167 |
> | TMI on random noise | .1792 | .4297 | .3738 | .0381 |
> | MIRROR-b | .2026 | .4533 | .3564 | .0613 |
>
> ---
>
> >[1] Goodfellow, Ian J., Jonathon Shlens, and Christian Szegedy. "Explaining and harnessing adversarial examples." arXiv preprint arXiv:1412.6572 (2014).
>
> >[2] Carlini, Nicholas, and David Wagner. "Towards evaluating the robustness of neural networks." 2017 IEEE symposium on security and privacy (sp). IEEE, 2017.
>
> >[3] Wang, Tianhao, Yuheng Zhang, and Ruoxi Jia. "Improving robustness to model inversion attacks via mutual information regularization." Proceedings of the AAAI Conference on Artificial Intelligence. Vol. 35. No. 13. 2021.
>
> >[4] Wen, Jing, Siu-Ming Yiu, and Lucas CK Hui. "Defending against model inversion attack by adversarial examples." 2021 IEEE International Conference on Cyber Security and Resilience (CSR). IEEE, 2021.
>
> >[5] Xu, Qian, Md Tanvir Arafin, and Gang Qu. "An approximate memory based defense against model inversion attacks to neural networks." IEEE Transactions on Emerging Topics in Computing 10.4 (2022): 1733-1745.
>
> >[6] Schonfeld, Edgar, Bernt Schiele, and Anna Khoreva. "A u-net based discriminator for generative adversarial networks." Proceedings of the IEEE/CVF conference on computer vision and pattern recognition. 2020.

---

> > ### Comment · Reviewer_wJqb · 2023-11-21
> >
> > Dear authors,
> >
> > I really appraciated your effort in improving the quality of the submission and in making it more useful for people interested in mitigation.
> >
> > Given these improvements, I am happy to increase my score to 6 and to side in favor of acceptance.

---

### Official Review · Reviewer_Xyhs · 2023-10-31

**Soundness:** 2 fair
**Presentation:** 2 fair
**Contribution:** 2 fair
**Rating:** 5
**Confidence:** 5

**Summary:**

This paper tackles the problem of black-box model inversion. The objective is to obtain the input data sample (or its surrogates) corresponding to a given prediction. While it is an undesirable scenario, it is important to make the attacks stronger and more pragmatic to be able to construct better countermeasures.

Previous methods on black-box model inversion are slow and do not often correspond to data samples that are specific to the given prediction.

This paper claims to address these issues effectively by learning a surrogate StyleGAN generator, followed by transforming the label prediction vectors onto the latent space of the generator, to be used for image generation.

**Strengths:**

1. The idea of using a surrogate generator and operating on the latent space of the same is good.

2. The paper positions itself well (describing the exact problem and gap in the literature).

3. Writing is fairly good (despite typos).

**Weaknesses:**

1. Dependence on a surrogate generator (and dataset).

2. Increased space-complexity.

3. Sloppy notations and incomplete math.

**Questions:**

1. The main problem I have about this method is that this demands training of a StyleGAN on a  dataset ``similar'' to that used in the predictor. This is not a fair assumption in my opinion. While the Authors do argue that the attacker can "leverage a pre trained StyleGAN network available on the Internet", it is a weak argument. How would the attacker know which dataset is ``similar" to the one used in the predictor? More grounding is needed in this respect, seems too handwavy currently.

2. Adding to the above point, the proposed method imposes an additional space constraint, in terms of the large stylegan that is to be trained.

3. Given the above two points, I do not see the comparisons to be fair as the proposed method has the luxury of using an additional full-blow generator network which the previous methods don't have. Therefore, I recommend that the Authors have to try modifying the other methods while they have access to a generator, albeit on a surrogate dataset.

4. The method seems very stylegan specific. Can a different GAN architecture be used? I am asking specifically because, in my experience, the latent spaces of other GANs are not as versatile as that of StyleGAN.

5. The description of the method has to be more formal. For instance, before Eq. 2, it is said that there is a loss function that is getting minimized. The optimization problem has to be stated neatly (using formal math).

6. A lot of mathematical details are missing - What is the expectation over, in Eq. 2 and 3? As I understand, they both are over two different distributions but are not mentioned.

7. While the Authors show a few images when there is a distributional shift between the surrogate model and the predictor, it is very minor and insignificant in my opinion. Both the datasets are facial images. What happens if you take a stylegan trained on cars dataset and use it for a predictor trained on human faces?

Overall, while the method is interesting, I have reservations about recommending it for acceptance given my concerns above. I shall wait to see other reviewers' comments and discussions with authors before making up my mind. Right now, I am leaning towards rejecting it.

---

> ### Author Response · Authors · 2023-11-21
> **Author Response to Reviewer Xyhs (1/2)**
>
> We thank the reviewer for the constructive feedback. We address each concern in the following.
>
> **W1+Q1. How would the attacker know which dataset is similar?**
>
> For a target ML service for users, it is trivial to know what kind of image they receive as input (face/medical/scenery/animal/...), the meaning of their output (similarity to celebrities/medical diagnosis/geolocation/...), and the characteristics of the output prediction (number of classes, meaning of each class) with little effort. Based on this information, the adversary can choose a pre-trained model or a dataset to use among numerous pre-trained models and datasets readily available on the Internet (e.g., Kaggle alone provides 268,605 datasets across 165 official subjects, including people, health, genetics, and business.). For instance, assume a scenario where the target service implements  geolocation estimation based on photos of scenery [10]. The service can predict which country or continent you are in based on the input photo. Accordingly, the attacker can decide to search for datasets containing landscape images (such as the Google landmarks dataset or InstaCities1M) or related pretrained generators for MI attacks.
>
> We emphasize that our adversary does not require additional knowledge compared to other prior MI attacks; our adversary leverages an auxiliary dataset that the adversaries that previous black- and white-box MI attacks [1-7] assumed.
>
> **W2+Q2. Imposes an additional space constraint, in terms of the large stylegan that is to be trained**
>
> We agree that training a StyleGAN network is a computational hurdle. However, building this StyleGAN network is a one-time procedure. After this preparation phase, MI attacks are completely offline while enabling high-fidelity input reconstruction for a large number of queries.
>
> **Q3. Unfair comparison; proposed method has the luxury of using additional full-blow generator network which the previous methods don't have**
>
> We note that all existing MI attacks [3-9] that utilize an image prior assume the same setting. Among the methods selected for baseline comparison, AMI is the only method that does not involve using a generator network, but still, it requires an auxiliary dataset to train the inverse mapping of the target model. One noticeable difference is that the auxiliary dataset is no longer needed for our TMI attacks that utilize an image prior, if there is a available pre-trained generator.  However, AMI does not support exploiting the pre-trained model.
>
> ---
>
> >[1] Fredrikson, Matthew, et al. "Privacy in pharmacogenetics: An End-to-End case study of personalized warfarin dosing." 23rd USENIX security symposium (USENIX Security 14). 2014.
>
> >[2] Yang, Ziqi, et al. "Neural network inversion in adversarial setting via background knowledge alignment." Proceedings of the 2019 ACM SIGSAC Conference on Computer and Communications Security. 2019.
>
> >[3] Zhang, Yuheng, et al. "The secret revealer: Generative model-inversion attacks against deep neural networks." Proceedings of the IEEE/CVF conference on computer vision and pattern recognition. 2020.
>
> >[4] Han, Gyojin, et al. "Reinforcement Learning-Based Black-Box Model Inversion Attacks." Proceedings of the IEEE/CVF Conference on Computer Vision and Pattern Recognition. 2023.
>
> >[5] Kahla, Mostafa, et al. "Label-only model inversion attacks via boundary repulsion." Proceedings of the IEEE/CVF Conference on Computer Vision and Pattern Recognition. 2022.
>
> >[6] An, Shengwei, et al. "Mirror: Model inversion for deep learning network with high fidelity." Proceedings of the 29th Network and Distributed System Security Symposium. 2022.
>
> >[7] Struppek, Lukas, et al. "Plug & Play Attacks: Towards Robust and Flexible Model Inversion Attacks." International Conference on Machine Learning. PMLR, 2022.
>
> >[8] Chen, Si, et al. "Knowledge-enriched distributional model inversion attacks." Proceedings of the IEEE/CVF international conference on computer vision. 2021.
>
> >[9] Wang, Kuan-Chieh, et al. "Variational model inversion attacks." Advances in Neural Information Processing Systems 34 (2021): 9706-9719.
>
> >[10] Weyand, Tobias, Ilya Kostrikov, and James Philbin. "Planet-photo geolocation with convolutional neural networks." Computer Vision–ECCV 2016: 14th European Conference, Amsterdam, The Netherlands, October 11-14, 2016, Proceedings, Part VIII 14. Springer International Publishing, 2016.

---

> ### Author Response · Authors · 2023-11-21
> **Author Response to Reviewer Xyhs (2/2)**
>
> **Q4. Can a different GAN architecture be used?**
>
> | | **Acc@1 ↑** | **Acc@5 ↑** | **F-dist↓** | **Cover ↑** |
> |-|:-:|:-:|:-:|:-:|
> | TMI | .3804 | .6255 | .2950 | .2067 |
> | TMI with UNet-GAN | .2201 | .4811 | .3254 | .2063 |
> | MIRROR-b | .2026 | .4533 | .3564 | .0613 |
>
> Yes, it is possible to utilize different GAN architectures since every GAN incorporates a latent space and a discriminator network. We conducted an additional experiment with UNet-GAN [11] pretrained on FFHQ and compared the results to the base experiment in the table above. For the surrogate model $f’$, we used the left-half of the UNet-GAN’s discriminator. Since UNet-GAN does not incorporate a mapping network, a custom mapping layer was trained from scratch. Observe that while the overall performance is slightly decreased compared to TMI with StyleGAN, it is still more effective than MIRROR-b. The slight drop in performance is largely due to the entangled latent space of UNet-GAN, where the latent space $\\mathcal{Z}$ is directly used without mapping it to an intermediate disentangled latent space $\\mathcal{W}$ in advance. We conclude that while StyleGAN is still the most effective image prior to be utilized, other GAN-based methods can generally benefit from our suggested approach of leveraging a new mapping layer and distilling the discriminator for a surrogate model. Therefore, the contribution of TMI is not only limited to the packaged attack method as a whole, but also the individual components that can be applied independently. The techniques from TMI might also be able to augment state-of-the-art diffusion models for MI attacks. For example, the mapping network can serve as a reliable encoder for latent diffusion models. We believe that designing an MI attack upon diffusion models is plausible, however, out of scope in this paper.
>
> **Q5+Q6. Incomplete math.**
>
> We appreciate the suggestion and will replace our math formulations in Eq2 and Eq3 with the formal equations as follows:
> - Eq2. $\\underset{\\theta}{\\arg\\min}\\,\\mathbb{E}\_{(w,x,\\hat{y})\\in\\mathcal{D}\_{gen}}\\left[ (w-m'\_\\theta(\\hat{y}))^2\\right]$
> - Eq3. $\\underset{\\theta}{\\arg\\min}\\,\\mathbb{E}\_{(w,x,\\hat{y})\\in\\mathcal{D}\_{gen}}\\left[ (\\hat{y}-f'\_\\theta(x))^2\\right]$
>
> **Q7. What happens if you take a stylegan trained on cars dataset and use it for a predictor trained on human faces?**
>
> Using an unrelated dataset will render TMI less effective, since its reconstruction domain is limited to the image manifold of the prior modeled in the generator. However, please note that as mentioned in **W1+Q1**, to the attacker’s perspective, it is probable to find a dataset or pretrained StyleGAN specific enough for TMI, given the abundance of public datasets and repositories. In addition, prior works on MI often use subsets from the target network’s trainset to train their inversion model, whereas our base evaluation for TMI utilize relevant, but different datasets. We believe that this adversary setting is an acceptable norm in the model inversion literature.
>
> ---
>
> >[11] Schonfeld, Edgar, Bernt Schiele, and Anna Khoreva. "A u-net based discriminator for generative adversarial networks." Proceedings of the IEEE/CVF conference on computer vision and pattern recognition. 2020.

---

> > ### Author Response · Authors · 2023-11-23
> > **Gentle Reminder for Discussion**
> >
> > Dear Reviewer,
> >
> > Thank you for your review once again. Above, we have clarifications and additional experiment results that address the issues raised in the review. It would be grateful if you could kindly let us know if there is anything else we can do for a better version of the paper.
> >
> > Thank you.

---

### Official Review · Reviewer_ZpeV · 2023-10-31

**Soundness:** 4 excellent
**Presentation:** 4 excellent
**Contribution:** 3 good
**Rating:** 8
**Confidence:** 4

**Summary:**

The authors seek to perform model inversion (creating the input that generated a prediction) by modifying a StyleGAN. Specifically, the StyleGAN mapping network is adapted to project a prediction vector into the GAN's latent space while the discriminator is adapted to be a surrogate model. Then to perform model inversion, a latent is created from the prediction vector and then optimized to generate an image that causes the surrogate model to emit a similar prediction vector. The authors are able to convincingly regenerate the input for a variety of  datasets.

**Strengths:**

I thought this paper was well-written, and the results were quite convincing. The method clearly outperforms other competing baselines, and is able to generate something resembling the model input. I appreciated that they chose two very different image datasets (celeba and chestxray).

 I think this could also be neat as an interpretability tool (e.g., to generate what images are on the border between two classes).

**Weaknesses:**

What is an example scenario where we do have access to the full prediction vector (e.g., the probabilities and not just the most likely class) but we (a) do not have the input and (b) we have limited queries to the model?

Regarding (b) I'm not sure query budget is necessarily the right metric (querying the surrogate model also takes time). It would be nice to see some scalability numbers on how long this method takes.

**Questions:**

How many queries do you have to perform to the surrogate model? I'd imagine that this can be just as expensive as querying the original model (if compute is the issue, I'm not sure "query budget" is the right metric to optimize).

How important is the surrogate model? If you used the original model instead (ignoring the query budget), how much better are your images?

Is this method specific to GANs (could you adapt one of the current off-the-shelf diffusion models for example).

---

> ### Author Response · Authors · 2023-11-21
> **Author Response to Reviewer ZpeV**
>
> We thank the reviewer for the constructive feedback. We address each concern in the following.
>
> **W1. Example scenarios**
>
> Consider an age classifier that gets a face-image query and classifies it into one of five classes that correspond to age groups, the adversary can populate a list of arbitrary prediction vectors that represent each class and then conduct TMI attacks to reconstruct the facial images by conducting TMI attacks. When this age classifier is hosted on an MLaaS cloud instance, the owner of this cloud instance should limit an excessive number of queries due to their computation and financial constraints. We explored this scenario in Section 5.3 (see Figure 4).
>
> We also described several example scenarios in Section 3 in which the adversary obtains prediction vectors. These scenarios include users posting prediction results on social media simply for entertainment (e.g. celebrity look-alike apps such as StarByFace show the look-alike percentage to celebrities), medical professionals sharing diagnosis predictions for educational or consultative purposes, split inference settings where the inference result is disclosed to different parties [1] or an untrusted MLaaS server. Prior studies have addressed countermeasures to protect inference privacy by obfuscating [2,3] or encrypting the input [4]. However, the prediction vector is left in plaintext format.
>
> **W2. How long does this method take + Q1. How many queries to the surrogate model?**
>
> | | **Query** | **Time (minutes)** |
> |-|-|-|
> | TMI | 5k (offline) | 2.38 |
> | AMI | 0 | $\\sim$0 |
> | MIRROR-b | 10k | 0.18 |
> | RLB-MI | 80k | 67 |
> | LO-MI | 25k | 1.91 |
> | P&P | 34k | 2.69 |
> | MIRROR-w | 160k | 72.67 |
>
> We compared the number of queries and the computation time per each attack in the table above. Please note that the required time is highly dependent to the baseline implementations and their own GAN prior. By default, TMI queries the surrogate model the same number of times to the update steps. As stated in A.1.2, we used $n=5000$ for our experiments. For other baselines, we assigned a default query budget that the respective attack assumed in their paper. We note that TMI with 5K updates exhibited the superior performance over all black-box baselines, as shown in Table 2. In the revised paper, we will include these experimental results. We emphasize that leveraging a surrogate model contributes to substantially decreasing the number of black-box queries directed at a target model (please refer to Section 5.3 for more details). Please note that the adversary can conduct MI attacks in an *offline* fashion by abusing this surrogate model, thus circumventing potential constraints associated with online queries. Consequently, the target service provider may remain unaware of whether they are under attack or not.
>
> **Q2. Replacing the surrogate model with the original (ignoring the query budget)**
> | | **Performance gain** |
> |:-:|:-:|
> | Acc@1 ↑ | .3408 → .8264 |
> | Acc@5 ↑ | .6255 → .9534 |
> | F-dist ↓ | .2950 → .1975 |
> | Cover ↑ | .2067 → .2619 |
>
> We appreciate this valuable suggestion. We conducted an additional experiment using the original model instead of the surrogate model, and the performance gains are listed in the table above. As expected, the performance is improved by an average of 63.7%, outperforming even the white-box attacks across all four metrics. This also indicates that retraining a mapping network can be applied for conducting  white-box TMI attacks to improve their reconstruction performance. We will include this result in A.2.1. Effect of the Surrogate Model.
>
> **Q3. Is this method specific to GANs?**
>
> Although our paper is focused on StyleGAN, the contribution of TMI is not only limited to the packaged attack method as a whole, but also the individual components that can be applied independently. For example, the mapping network might also serve as a reliable encoder for latent diffusion models. We believe that designing an MI method using a diffusion models is also promising. However, we consider exploring this direction out of scope for this paper.
>
> ---
>
> >[1] Zecheng He, Tianwei Zhang, and Ruby B Lee. Model inversion attacks against collaborative inference. In Proceedings of the 35th Annual Computer Security Applications Conference, pp. 148–162, 2019.
>
> >[2] Mireshghallah, Fatemehsadat, et al. "Shredder: Learning noise distributions to protect inference privacy." Proceedings of the Twenty-Fifth International Conference on Architectural Support for Programming Languages and Operating Systems. 2020.
>
> >[3] Liu, Qin, et al. "When deep learning meets steganography: Protecting inference privacy in the dark." IEEE INFOCOM 2022-IEEE Conference on Computer Communications. IEEE, 2022.
>
> >[4] Gu, Zhongshu, et al. "Privacy enhancing deep learning cloud service using a trusted execution environment." U.S. Patent No. 11,443,182. 13 Sep. 2022.

---

> > ### Comment · Reviewer_ZpeV · 2023-11-21
> > **Response**
> >
> > I thank the reviewers for their response. I keep my positive score of 8 (accept)

---

### Official Review · Reviewer_6NZM · 2023-11-02

**Soundness:** 3 good
**Presentation:** 3 good
**Contribution:** 2 fair
**Rating:** 5
**Confidence:** 3

**Summary:**

The paper presents a new model inversion (MI) attack leveraging StyleGAN as an image prior in a blackbox fashion -- i.e., without requiring access to the model's weights to compute gradients through it. The goal of a MI attack is to successfully reconstruct the input image to a classifier, based only on the predictions from it, as a result of which, MI can be used to recover samples from the training dataset, leaking potentially sensitive information (personal details, health scans, face images etc.). Developing better MI attacks are useful so they can be defended against more effectively for sensitive applications.

While there has been significant progress in white-box MI attacks, black box methods still have a way to go because they are harder, and typically require several hundreds of thousands of queries for any reasonable inversion. Further, some existing methods tend to produce generic class-representative images, at the cost of intra-class differences, which undermines the goal of reproducing privacy preserving attacks.

This paper presents a _Targeted Model Inversion (TMI)_  attack, that modifies StyleGAN's latent mapper ($g: \mathcal{Z} \rightarrow \mathcal{W}$), that goes from a random prior distribution (like Gaussian) to the W space of StyleGAN. The modified mapper, $m( . )$ directly predicts the $w \in \mathcal{W}$, which is then passed to the generator to obtain the image; this is trained by optimizing in the W space. A surrogate model, $f'$ is obtained by fine-tuning adapting the pre-trained discriminator to mimic the original model, while the images are sampled from the pre-trained StyleGAN.

**Strengths:**

MI is an important problem to study, especially considering the potential for damage that can be caused with sensitive data. The paper addresses blackbox MI attacks, which is a more practical scenario where the attacker only has access to the model via an API call.
* The use of StyleGAN as an image prior, and the discriminator as the surrogate makes intuitive sense, and exploiting it for MI is a realistic scenario.
* Use of a general, strong prior like StyleGAN also produces sufficient intra-class diversity  -- the empirical results also show that in terms of diversity TMI outperforms other whitebox methods, significantly which is encouraging.
* Evaluations are convincing, and the different metrics considered demonstrate the superiority of TMI over blackbox and whitebox methods.
* The data efficiency for similar or better accuracy in MI attack over baselines is promising

**Weaknesses:**

* **Image Prior** An unacknowledged weakness of the paper is the generality of the approach to a broader set of application domains. By choosing a StyleGAN prior, the applicability of the current method becomes restricted to the domains on which (or domains related to) to the StyleGAN's training distributions. The evaluations, and experiments -- while impressive are of less impactful in my opinion.
* Two potential mitigation strategies come to mind -- (a) in order to work with more SoTA foundation models like StyleGAN-XL, the current approach will require to work with conditional generative models, which can make it much more potent and realistic; or (b) Leverage stronger inversion techniques on _unconditional_ StyleGAN that are able to invert arbitrary OOD images using pre-trained styleGAN as well -- for example
	* GAN inversion for out-of-range images with geometric transformations, CVPR'21
	* Image2stylegan++: How to edit the embedded images?, CVPR'21
	* Improved StyleGAN-v2 based Inversion for Out-of-Distribution Images, ICML'22
* The examples of images that are "significant" deviation from the original dataset, unfortunately are not that OOD. It's well known that most encoders (including pSp or e4e ) do a reasonable job of inverting these paintings and other closely related domains. So i think a more accurate test of TMI will be to use a much more generic dataset, which will likely fail.
* The feature distance measure appears to be poorly correlated to image quality -- i think the metric maybe misleading since an approach like AMI, which arguably fails on most of the evaluations conducted, has a reasonable F-dist score comparable or bette than some of the other baselines which are clearly better.

**Questions:**

In addition to some of my comments above --
* Can the ablations on query budget be done for one of the baselines as well to see how well (or badly ) they behave as the budget decreases?
* How does the surrogate perform on the original dataset? is it an actual surrogate in the sense that it come close to the original model's performance? this might be an interesting ablation as well.
* A (non-technical) question -- is the use of gendered language for describing the attacker common?

>"..where the adversary _(Eve)_ is able to.."

>".. *She* uses $D_{aux}$ to train their *her* StyleGAN network"..

I found that a bit odd but i am not closely familiar with how this is done in security and safety research so I will defer to the authors.

---

> ### Author Response · Authors · 2023-11-17
> **Author Response to Reviewer 6NZM's (1/2)**
>
> We thank the reviewer for the constructive feedback. We address each concern in the following.
>
> **Q1. Ablations on query budget**
>
> |  | **Query budget** | **Acc@1↑** | **Acc@5↑** | **F-dist↓** | **Cover↑** |
> |-|-:|:-:|:-:|:-:|:-:|
> | **P&P** | 5k&nbsp;&nbsp;&nbsp;&nbsp; | .101 | .284 | .393 | .041 |
> |  | 10k&nbsp;&nbsp;&nbsp;&nbsp; | .104 | .299 | .390 | .046 |
> |  | 50k&nbsp;&nbsp;&nbsp;&nbsp; | .156 | .364 | .373 | .060 |
> |  | 100k&nbsp;&nbsp;&nbsp;&nbsp; | .153 | .388 | .365 | .048 |
> |  |  |  |  |  |  |
> | **MIRROR-w** | 5k&nbsp;&nbsp;&nbsp;&nbsp; | 0 | .013 | .485 | .020 |
> |  | 10k&nbsp;&nbsp;&nbsp;&nbsp; | .026 | .071 | .460 | .031 |
> |  | 50k&nbsp;&nbsp;&nbsp;&nbsp; | .259 | .478 | .365 | .059 |
> |  | 100k&nbsp;&nbsp;&nbsp;&nbsp; | .354 | .609 | .341 | .062 |
> |  |  |  |  |  |  |
> | **TMI** | 5k&nbsp;&nbsp;&nbsp;&nbsp; | .232 | .448 | .340 | .121 |
> |  | 10k&nbsp;&nbsp;&nbsp;&nbsp; | .274 | .540 | .318 | .144 |
> |  | 50k&nbsp;&nbsp;&nbsp;&nbsp; | .358 | .646 | .286 | .182 |
> |  | 100k&nbsp;&nbsp;&nbsp;&nbsp; | .341 | .626 | .295 | .207 |
>
> We conducted additional experiments to assess the performance of the baseline methods across varying query budgets (5, 10, 50, 100K). Please note that the query budget denotes the total number of queries allowed to attack every label (i.e., 530 attacks in case of FaceScrub). We included the experimental results of two best-performing baseline methods (i.e., P&P and MIRROR-w) and our TMI attack in the table above. The graphical representations of the performance of all baseline methods can be found ***[here](https://drive.google.com/file/d/1q_2EXRYds6srSQQ2gETbRreVYIjDJo1n)***. As the table above shows, the overall MI performance increases as the number of queries increases. TMI consistently outperforms the baseline attacks across different query budgets. We also note that TMI holds a distinctive advantage of requiring zero queries to a target model after the preparation phase, which significantly facilitates the reconstruction of inputs for a large number of prediction vectors.
>
> During the experiments, we identified an error in the query count of LO-MI in Table 1. For each attack attempt, LO-MI requires 25k queries, instead of 2k. This will be corrected in the final version. We confirm that this erroneous typo does not impact the overall conclusion of our study since we overstated the performance of the baseline attack (i.e., LO-MI). Correcting this value highlights the superior performance of TMI attacks.
>
> **Q2. Surrogate model’s performance on the original dataset?**
>
> We measured the accuracy of the TMI surrogate models using the test splits of the FaceScrub, CelebA, and PadChest datasets. The table below describes the performance metrics of both original ($f$) and surrogate ($f’$) models. In the revised paper, we will include these experimental results.
>
> | **Dataset** | **Architecture** | **$f$ Acc.** | **$f'$ Acc.** |
> |-|-|:-:|:-:|
> | FaceScrub | ResNeSt-101 | 0.961 | 0.605 |
> | | DenseNet-169 | 0.954 | 0.588 |
> | | MobileNet-v3 | 0.937 | 0.542 |
> | CelebA | ResNeSt-101 | 0.877 | 0.580 |
> | | DenseNet-169 | 0.849 | 0.582 |
> | | MobileNet-v3 | 0.792 | 0.552 |
> | PadChest | ResNeSt-101 | 0.730 | 0.708 |
> | | DenseNet-169 | 0.722 | 0.700 |
> | | MobileNet-v3 | 0.732 | 0.686 |
>
> **Q3. Use of gendered language for describing the attacker**
>
> We will replace those words with gender-neutral terms. We note that prior studies in the security and privacy domain have often employed gender-specific terms when referring to the adversary.

---

> ### Author Response · Authors · 2023-11-17
> **Author Response to Reviewer 6NZM's (2/2)**
>
> **W1. Applicability of the current method restricted to domains related to the StyleGAN's training distributions**
>
> We acknowledge that TMI necessitates a StyleGAN prior model. However, all previous MI studies [1-9] have assumed access to an auxiliary dataset of which underlying distribution is either the same or slightly different from the distribution of the target dataset. Therefore, we proposed a new MI method that harnesses this auxiliary dataset in a black-box manner, achieving superior MI performance over the SoTA methods.
>
> In addition, we note that our TMI evaluation spans two distinct application domains, namely chest X-ray and facial datasets. This suggests that, given an appropriate StyleGAN prior, TMI is applicable to diverse classification tasks for MI. Given the abundance of pre-trained models and datasets readily available on the internet (e.g., Kaggle alone provides 17,678 image datasets across 164 official subjects, including people, healthcare, genetics, and business.), we believe that the adversary is capable of selecting their preferred StyleGAN prior for their needs. Even when there is no available dataset or pretrained StyleGAN, the adversary still has an option to collect their own images and train a StyleGAN specific to their need. Image search or sharing services (such as Google Images, Flickr, and Pinterest) allow attackers to access a large volume of image data.
>
> **W2. Explore leveraging advanced GAN techniques**
>
> We will discuss potential improvements by integrating SoTA models and the connection between GAN inversion to MI attacks. GAN inversion typically requires an original image to find a suitable latent point, whereas our TMI attacks do not assume white-box access to the target model or access to the original image. Leveraging a prediction output without the original image, TMI performs the best for identifying the optimal latent point that contributes to reconstructing the input image among the SoTA MI methods.
>
> **W3. more accurate test of TMI will be to use a much more generic dataset, which will likely fail**
>
> Using a much more generic dataset will render TMI less effective, since its reconstruction domain depends on the image manifold of the prior modeled in the generator. However, please note that as mentioned in **W1**, to the attacker’s perspective, it is probable to find a dataset or pretrained StyleGAN specific enough for TMI, given the abundance of public datasets and repositories.
> In addition, prior works on MI often use subsets from the target network’s trainset to train their inversion model, whereas our base evaluation for TMI utilize relevant, but different datasets. We believe that this adversary setting is an acceptable norm in the model inversion literatures. Also note that TMI assumes a weaker black-box adversary with a limited number of input queries.
>
> **W4. Feature distance poorly correlated to image quality**
>
> We appreciate this valuable insight and agree that feature distance alone is insufficient to demonstrate reconstruction performance. Therefore, we used four metrics to comprehensively measure the MI performance of TMI and the other baselines.  We will discuss the downsides of using feature distance alone in measuring MI performance.
>
> ---
>
> >[1] Fredrikson, Matthew, et al. "Privacy in pharmacogenetics: An End-to-End case study of personalized warfarin dosing." 23rd USENIX security symposium (USENIX Security 14). 2014.
>
> >[2] Yang, Ziqi, et al. "Neural network inversion in adversarial setting via background knowledge alignment." Proceedings of the 2019 ACM SIGSAC Conference on Computer and Communications Security. 2019.
>
> >[3] Zhang, Yuheng, et al. "The secret revealer: Generative model-inversion attacks against deep neural networks." Proceedings of the IEEE/CVF conference on computer vision and pattern recognition. 2020.
>
> >[4] Han, Gyojin, et al. "Reinforcement Learning-Based Black-Box Model Inversion Attacks." Proceedings of the IEEE/CVF Conference on Computer Vision and Pattern Recognition. 2023.
>
> >[5] Kahla, Mostafa, et al. "Label-only model inversion attacks via boundary repulsion." Proceedings of the IEEE/CVF Conference on Computer Vision and Pattern Recognition. 2022.
>
> >[6] An, Shengwei, et al. "Mirror: Model inversion for deep learning network with high fidelity." Proceedings of the 29th Network and Distributed System Security Symposium. 2022.
>
> >[7] Struppek, Lukas, et al. "Plug & Play Attacks: Towards Robust and Flexible Model Inversion Attacks." International Conference on Machine Learning. PMLR, 2022.
>
> >[8] Chen, Si, et al. "Knowledge-enriched distributional model inversion attacks." Proceedings of the IEEE/CVF international conference on computer vision. 2021.
>
> >[9] Wang, Kuan-Chieh, et al. "Variational model inversion attacks." Advances in Neural Information Processing Systems 34 (2021): 9706-9719.

---

> > ### Comment · Reviewer_6NZM · 2023-11-21
> >
> > I think the authors for the response and the additional experiments. I have a few observations
> > * TMI is clearly very effective even under small query budgets, these ablations are impressive.
> > * Interesting to see the surrogates performance in a black box attack is reasonable -- which suggests the power of a powerful prior like StyleGAN.
> >
> > *StyleGAN and related domains*
> > I appreciate the authors response here that it is in fact easy to train StyleGANs on any domain since there are so publicly available datasets. However, I find it a stretch to infer from this that the attack will be successful on any domain -- true that the paper has tested on two distinct domains -- but these represent a very tiny subset of the wide variety of classifiers typically encountered.
> >
> > Another reviewer has also raised an important concern that I share -- determining which domain is in-distribution to choose an appropriate StyleGAN for the attack. Typical large scale classifiers are often pre-trained on massive datasets, following which they are fine-tuned on downstream tasks (very much like how the current paper also does). Its well known that these types of models are very good at generalizing to unseen domains. In a blackbox setup, I don't see how it is trivial to determine this distributional match to the StyleGAN.
> >
> > A way to mitigate this (as I have suggested in my review) is to show that the prior is not as important using far OOD images or on tasks that are larger scale like ImageNet, both of which the current paper does not do -- and which likely requires a non-trivial amount of change. Another limitation is that increasingly generative models are moving away from (unconditional) StyleGAN architectures, which again limits the applicability of the method.
> >
> > The paper has a lot of promise, but given its empirical nature and weaknesses i have outlined, I see this as a borderline paper and I am leaning on retaining my original score.

---

### Meta-Review · Area_Chair_mcjG · 2023-12-07

**Metareview:**

This paper focuses on the black box model inversion attack, which is a challenging problem with respect to high-resolution inversion and a limited query budget. The authors adopt StyleGAN as a surrogate model to produce high-quality pseudo images. Extensive experiments demonstrate the effectiveness of the proposed approach.

Strengths:

(1)   Black box model inversion is an important and challenging problem and has many practical applications. The motivation for this work is clear.

(2)   The paper is well-written and the idea is easy to follow.

(3)   The experiments are promising to demonstrate the efficacy of the proposed approach.

Weaknesses:

(1)   The biggest limitation I find of the work is the highly restrictive assumptions made to demonstrate the contribution. The proposed method highly depends on a well-trained StyleGAN model. It's unclear whether this approach can work if the prior relaxed to be more OOD than those considered in the paper.

(2) The current model inversion attack has been scaled to a large-scale dataset (imagenet level). The experiments in the paper are slightly simpler today, it would have been useful to see evidence of how the proposed method might scale to imageNet-level classifiers or other large-scale datasets.

After the authors' response and discussion with reviewers,  there still exist several concerns that are not well-solved. Although the black box model inversion attack in this work show shows good performance, the current experiments fail to demonstrate their proposed approach can tackle more diverse, OOD and large-scale datasets. Therefore, I recommend rejection.

**Justification For Why Not Higher Score:**

N/A

**Justification For Why Not Lower Score:**

N/A

---

### Decision · Program_Chairs · 2024-01-16

Reject